# Application of the Pulse Infrared Thermography Method for Nondestructive Evaluation of Composite Aircraft Adhesive Joints

**DOI:** 10.3390/ma14030533

**Published:** 2021-01-22

**Authors:** Tomáš Kostroun, Milan Dvořák

**Affiliations:** 1Department of Aerospace Engineering, Faculty of Mechanical Engineering, Czech Technical University in Prague, Technická 4, 160 00 Praha 6, Czech Republic; 2Department of Mechanics, Biomechanics and Mechatronics, Faculty of Mechanical Engineering, Czech Technical University in Prague, Technická 4, 160 00 Praha 6, Czech Republic; milan.dvorak@fs.cvut.cz

**Keywords:** NDE, infrared thermography, Infrared Nondestructive Testing, composite, CFRP

## Abstract

In this article, we examine the possibility of using active infrared thermography as a nontraditional, nondestructive evaluation method (NDE) for the testing of adhesive joints. Attention was focused on the load-bearing wing structure and related structural joints, specifically the adhesive joints of the wing spar caps and the skins on the wing demonstrator of a small sport aircraft made mainly of a carbon composite. The Pulse Thermography (PT) method, using flash lamps for optical excitation, was tested. The Modified Differential Absolute Contrast (MDAC) method was used to process the measured data to reduce the effect of the heat source’s inhomogeneity and surface emissivity. This method demonstrated a very high ability to detect defects in the adhesive joints. The achieved results are easy to interpret and use for both qualitative and quantitative evaluation of the adhesive joints of thin composite parts.

## 1. Introduction

This article describes our research into the viability of the quality control of a strength-critical adhesive joint on an all-composite aircraft and follows up on the work described in [1]. At present, the market for small high-performance sports aircraft, which are mostly built of composite materials, is developing rapidly. These aircraft are designed and operated in the Light Sport Aircraft (LSA) category. The LSA regulations require a certificated strength test of the prototype. Other aircraft in the production series are then operated using periodic inspections, but these are only done by sight. None of the known Nondestructive Evaluation (NDE) methods are mandatory and are also not used for financial reasons. The variance in production quality, which is typical for composite structures, is covered by the special safety factor of 1.20–1.50 [2]. The basic safety factor of 1.5 is then multiplied by this special factor. It is clear that the use of a simple and fast control method would make it possible to safely use the lower limit of the increasing coefficient and thus safely operate the optimized lightweight aircraft structure. Another factor is that LSA-category planes are not monitored while in service from the point of view of their fatigue life; the general assumption is that the composite aircraft is effectively obsolete before it reaches the technical end of its life. Due to the nature of LSA production, which is small-scale or a piece type with a wide range of designs, the pressure on the price of such NDE solutions is high, as it is inevitably passed on to the sale price. As previously mentioned, the current trend is for all-composite aircraft design. Typically, these aircrafts’ constructions are based on precured composite parts. They are made by a contact hand lamination method, using vacuum-assisted resin transfer methods or by means of prepreg technology using an autoclave [3,4]. These assemblies, for example the fuselage halves, the wing skins, wing spars, etc., are typically connected using adhesive joints. Both the composites and the adhesive joints bring new issues and problems to the subject of NDE testing. 

The aim of this experimental work was to evaluate the bonding quality of small composite aircraft wing adhesive joints. The all-composite wing was assembled from precured carbon-fiber-reinforced plastic (CFRP) composite parts, using a two-component epoxy adhesive system. This work was focused on the critical adhesive joints of the wing skin to the spars and ribs. 

The composite design of LSA-category aircraft creates new demands for the detection capabilities of NDE methods. Although in general aviation, the thickness of glued composite parts can be within a range of millimeters to tens of millimeters [5], in the LSA category, composite parts are often very thin, with thicknesses in the tenths of millimeters [3,4].

The problem is an interesting and challenging one due to the combined requirements involved in evaluating the quality of the adhesive joints of the composite parts. In addition to the currently developed methods, such as Fiber optic Bragg Grating (FBG) sensors integrated directly in the adhesive joint layer [6], only some Nondestructive Evaluation methods are usable. NDE methods are generally used to detect hidden defects in a material, such as air bubbles, poorly impregnated composite fabric or defects caused by mechanical impact or excessive load, without damaging the part being tested or affecting its working properties. Traditional NDE methods include eddy current testing, acoustic emission, bond testing, X-ray and ultrasonic testing, infrared testing and others [7,8,9].

One of the methods that can potentially provide a solution for NDE testing of the glued joints of small sports aircraft structures is a group of methods using active infrared thermography. Active infrared thermography is an inspection technique that requires an external energy source to create a temperature difference between defective and nondefective areas of the specimen during testing. There are several methods of providing a heat source from which to choose, such as excitation by light, laser, hot air, ultrasonic or microwave radiation [9,10,11]. 

According to available sources, the Pulse Active Infrared Thermography Nondestructive Testing (PT IRNDT) method seems to be a suitable solution for solving this problem. This method uses a short light pulse generated by a flash lamp or a halogen light. The PT IRNDT method allows for the rapid inspection of specific areas of the tested object with a direct 2D graphical output. This provides relatively easy-to-interpret results. It is suitable for the finding of flaws and voids located close to the surface [12,13,14,15,16,17,18,19,20,21,22,23,24]. 

In the case of the tests performed on the composite wing, defects in the adhesive joints were expected to be found at a depth of 0.5–1.8 mm below the tested wing skin surface. These would be difficult to detect using different standard NDE methods, such as ultrasonic testing or X-rays. 

The first method described assumes the use of either an attenuation (through-transmission) method or a reflection (pulse-echo) method. The attenuation method cannot be used here because adhesive joints are not accessible along their entire length from both sides. Due to the small sizes of the measured thicknesses, the use of the reflection method is only possible when using an immersion technique, which requires immersion of the part in a water bath, or the use of a water-squirt configuration (Water Jet Device), which is more suitable for the pass-through method. The immersion technique requires the use of a scanning device, or even a large vessel, into which the test part must be immersed. With the size of the construction being tested (a wingspan larger than 7 m), such a device is not commonly available. In addition, even when using a water-squirt configuration instead of the immersion of the complete part, the resulting moisture is generally problematic for composites [10,11].

The second potential method is with X-ray, but, again, the major problem is the size of the part and accessibility to the tested area from both sides of an adhesive joint. In addition, this method evaluates the change of the thickness of the inspected part of the structure so it will not detect the disbonding of the joint if the adherends still overlap.

## 2. Methods

### 2.1. Experimental Composite Wing Specimen

The composite wing being tested came from a two-seater aircraft designed for the LSA category. The maximum take-off mass of the aircraft is 600 kg, with a wingspan of about 7.2 m and a length of 7.5 m. The aircraft is powered by a piston engine. This category of aircraft is characterized by the use of extremely thin-walled structures. Commonly used NDE methods are not suitable for this task because of their lower sensitivity.

#### 2.1.1. Composite Wing Description

The test sample was the right half of the wing of a small composite aircraft demonstrator. The wing was used for the development of static structural strength testing. Its structure consisted of spars and ribs made from CFRP. The wing skins were made of a sandwich structure consisting of a carbon sheet and a foam core. The outer and inner skins were made from one layer of CFRP fabric, using TeXtreme^®^ 100 material (Oxeon AB, Boras, Sweden), with a nominal layer thickness of 0.1 mm. Wing skins were glued to the load-bearing structure using the HexBond^®^ EA9394 epoxy adhesive (Hexcel Corporation, Stamford, CT, USA). Bonding was done by applying a thin layer of adhesive to the wing skin and a thicker layer of glue in the shape of a “snake” on the wing spar caps and ribs. Subsequent curing was done in the assembly jig. The maximum thickness of the adhesive layer should be up to 1.5 mm. Figure 1 shows a system drawing of the upper half of the wing (the lower half is similar), with the areas of interest marked for testing. The scheme of the configuration of the bonded joints in these areas is pictured in Figure 2.

#### 2.1.2. Material Properties

Within this work, a determination of the basic thermal properties of both HexBond^®^ EA9394 and TeXtreme^®^ 100, in the direction perpendicular to their surfaces, was performed. The density *ρ*, thermal diffusivity *α* and specific heat capacity *c_p_* were determined experimentally. From the data obtained, the values of thermal conductivity λ and thermal effusivity *e* were subsequently determined by calculation.

Experimental specimens from the examined materials were made with an outer diameter of *D* = 48 mm. The thicknesses of the specimens were *z* = 1.745 mm for the EA9394 material and *z* = 2.510 mm for the TeXtreme^®^ 100 material. The composite lay-up of the second specimen was as follows: [0°, 90°]_6S_. The thickness of the reference specimen, which was made of aluminum alloy 6061 T6, was *z* = 1.240 mm.

The densities of the materials being examined were determined by measuring the difference between the weights of the specimens in air and the weights of the same specimens when immersed in distilled water according to Archimedes’ law. The configuration of the test is shown in Figure 3. The resulting densities were calculated according to Equation (1)
(1)$$\rho =\frac{m}{\Delta m_{w}}\left(\rho_{w}-\rho_{a}\right)+\rho_{a}$$
where *m* is the weight of the sample weighed in air, Δ*m_w_* is the difference between the weights of the sample measured in air and the samples immersed in water, *ρ_w_* is the density of water (*ρ_w_* = 998 kg·m^–3^ at 20 °C) and *ρ_a_* is the density of air (*ρ_a_* = 1.2 kg·m^–3^). The resulting density values for the individual materials are given in Table 1.

The thermal diffusivity, *α*, of these materials was measured on the basis of the thermal curve of the surface of a thin specimen, after excitation by a short thermal pulse. Excitation was performed from the other side of the specimen using a flash lamp. This measuring technique is based on the ASTM E 1461 standard [10]. The value of thermal diffusivity is calculated from the thickness of the sample *z* and the time *t* (measured from the excitation moment), when the temperature increase, Δ*T*, on the measured surface reaches a certain percentage of the maximum surface temperature increase Δ*T_max_*. The surface temperature of the specimen was measured using an Infrared (IR) camera and determined as the average value from a circular area in the center of the specimen, which had a diameter of 10 mm. To ensure the same surface emissivity for all specimens, a thin layer of paint with a defined emissivity *ε* (Therma Spray 800 with *ε* = 0.96) was applied to both sides of the specimen. For each specimen, 5 measurements were performed and evaluated, and the resulting value of the thermal diffusivity *α* is their average.

Equation (2) [25] was used to determine the thermal diffusivity *α*_0.5_ in time *t*_0.5_:(2)$$\alpha =0.13879\frac{z^{2}}{t_{0.5}}$$
where *z* is the specimen thickness, and *t*_0.5_ is the time taken for the temperature to rise to 50% Δ*T_max_*.

Since the above equation is based on the simplified assumption that no heat losses occur during the test, a correction for these losses needs to be done. One possibility is to use the correction reported by Clark and Taylor [25,26], which is based on the ratio of times *t*_0.25_ and *t*_0.75_, when the temperature rise reaches 25% and 75% of the maximum temperature rise, respectively. The correction factor *K_R_* is then calculated according to Equation (3):(3)$$K_{R}=-0.3461467+0.361578\frac{t_{0.75}}{t_{0.25}}-0.06520543{\left(\frac{t_{0.75}}{t_{0.25}}\right)}^{2}$$

The corrected value of the thermal diffusivity *α_corr_* is then (4):(4)$$\alpha_{corr}=\frac{\alpha_{0.5\;}K_{R}}{0.13885}$$

A measuring device was assembled for the purpose of this experimental work. Its scheme can be seen in Figure 4. The device consisted of a flash lamp with a reflector, a specimen holder and an IR camera to record the temperatures.

The resulting values of thermal diffusivity are given in Table 2 and Table 3. Figure 5a,b shows the cooling curves for the EA9394 material specimen (left) and the TeXtreme^®^ 100 material specimen (right).

The standard deviation value given in Table 2 and Table 3 is the sample standard deviation from the measured (calculated) values defined by Equation (5):(5)$$s=\sqrt{\frac{\sum {\left(x_{i}-x_{a}\right)}^{2}}{n-1}}$$
where *x_i_* is the measured (calculated) value from the *i*-th measurement, *x_a_* is the arithmetic mean of the values from the measured (calculated) data and *n* is the number of measurements.

The determination of the specific heat capacity of the material is based on the assumption that the supplied heat *Q* per unit area *A* is manifested by an increase in temperature depending on the density of the material *ρ*, the specific heat capacity *c_p_* and the specimen thickness *z*, as shown in Equation (6).
(6)$$\frac{Q}{A}=\Delta T\times c_{p}\times z\times \rho$$

If we perform these measurements under the same conditions (temperature, heat input) for different specimens, where for one reference specimen (index R), all parameters are known, and for the others, the only unknown value is the specific heat capacity, it is possible to determine unknown specific heat capacity *c_p_* according to Equation (7) [27]:(7)$$c_{p}=\frac{\Delta T_{R}}{\Delta T}\frac{z_{R}\times \rho_{R}}{z\times \rho }c_{pR}$$

The specimen was made of the aluminum alloy 6061 T6 with a thickness of *z_R_* = 1.240 mm, density of *ρ_R_* = 2687 kg·m^−3^ and specific heat capacity of *c_p_* = 896 J·kg^−1^·K^−1^ was used as a reference standard. This measurement was performed on the same equipment, with the same specimens and under the same conditions as the thermal diffusivity measurement. The individual values of the temperature increases are given in Table 4. The specific heat capacity values were subsequently determined from the average values of the temperature increase parameter.

The values of thermal conductivity *λ* and thermal effusivity *e* were calculated from the determined thermal properties according to Equations (8) and (9):(8)$$\lambda =\;\alpha \times \rho \times c_{p}$$
(9)$$e=\;\sqrt{\lambda \times \rho \times c_{p}}$$

A summary of the determined and calculated thermal properties is given in Table 5.

### 2.2. PT Experimental Method Description

#### 2.2.1. PT Theory

The PT method is based on the principle of heating a sample from one side with a short thermal pulse (for example, a halogen light or a flash lamp) and the subsequent monitoring of the cooling curve at each point of the surface using an IR thermal camera. By sending a pulse, the heat wave begins to propagate through the material. The surface cools due to heat wave propagation (conduction) into the depth of the material as well as due to convection and radiation losses. If beneath the surface there is a defect with a different thermal effusivity to that of the base material (delamination, cavity or void in an adhesive joint), the heat wave will be reflected back to the surface and the cooling process will change at this point. This behavior of the surface cooling curves is demonstrated in Figure 6. Defects that occur at a greater depth will appear on the thermogram with a time delay [12,16]. The time *t* required to manifest the temperature deviation is a function of the depth of the defect *z* and the thermal diffusivity *α* according to the relation (10) [12]
(10)$$t\;\propto \;\frac{z^{2}}{\alpha }$$

Figure 7 shows an example of the time evolution of thermograms for the different moments after the excitation pulse. The figure shows a time-sequential drawing of the deeper layers of the adhesive joint. At time *t* = 5 s, the poor quality of the joint can be seen as resulting from inadequate technology (the adhesive bead was not compressed and spread sufficiently). At the same time, it can be seen that due to lateral diffusion, thermograms lose their sharpness with increasing time.

The disadvantage of this method is a sensitivity to the unevenness of the heat source and the distribution of emissivity on the surface. This can be partially eliminated by subsequent postprocessing.

#### 2.2.2. PT Method Verification

For the purpose of verifying the adhesive joints testing method, and for the setting up of the measuring device, a reference gauge was produced. This gauge corresponds in its composition to the point of the adhesive joints on the wing (Layers 1, 3, 4 in Figure 2). The individual thicknesses represent the depths of the occurrence of the defects of the adhesive joint formed within the adhesive-air interface. The total thickness of the gauge is cut in a range of *z* = 0.25 mm (skin alone) to *z* = 2.3 mm (skin + adhesive). With the assumed maximum thickness of the adhesive layer in the adhesive joint of approx. 1.5 mm, this larger gauge thickness should represent a correctly glued joint. Due to the homogenization of the emissivity of the surface, the gauge was sprayed with paint with a defined emissivity of *ε* = 0.96. The dimensions of the reference gauge are shown in Figure 8.

Figure 9 shows the sequence of thermograms at five different time points *t* after excitation. It represents the temperature distribution on the surface of the gauge. The warmest spot is represented by a white color; the coldest by black, with each image in the sequence being normalized to achieve the maximum dynamic range. This figure clearly shows the gradual delineation of individual thicknesses and the gradual blurring of the boundaries between the individual steps of the gauge due to lateral diffusion. Local deviations in temperature distribution within the individual steps of the gauge are caused by imperfections in its production (deviations from the optimal thickness and the occurrence of air voids in the adhesive).

The following graph in Figure 10 shows the course of temperature distribution along the longitudinal axis of the gauge for the same time steps as in the previous thermograms. The temperature curves are again normalized separately for each curve. From the above thermograms and temperature distribution curves, it is clear that the performed measurement confirms an ability to detect defects in the adhesive joint within the entire range of expected depths.

Figure 11a,b shows the time course of the cooling curves measured at the centers of the individual steps of the reference gauge. The first image has a linear timeline, while the second image is logarithmic. The second figure clearly shows a turning point in the temperature decrease at time *t* = 0.05–0.10 s after excitation, which is caused by the different thermal diffusivity of the skin and of the adhesive material. Both graphs show a time-varying deviation of the individual cooling curves from the curve, representing a correctly made joint (*z* = 2.3 mm).

Since the resulting thermograms are normalized to the temperature range for a given time, the following graph in Figure 12 shows the temperature profile normalized to the temperature range from the highest temperature (curve for *z* = 0.25 mm) to the lowest temperature (curve for *z* = 2.3 mm) at each measurement point. This graph shows the dimensionless contrasts within the data obtained at a given time.

#### 2.2.3. Differential Absolute Contrast (DAC) Evaluation

A Modified Differential Absolute Contrast (DAC) method was used to process the measured data. This allows partial elimination of heat source unevenness, such as reflections from the surroundings (e.g., IR camera, flash lamps) and emissivity distribution on the surface. The DAC method compares the temperature of the tested place containing the defect, with the theoretical value of the temperature if there were no defect in the place being tested. This theoretical temperature is calculated on the basis of the 1D form of the Fourier equation of heat conduction in a semi-infinite medium from the measured temperature, at a point in time where the temperature defect does not manifest itself [12]. The standard DAC method works with the temperature at that point in time just before the manifestation of the defect, which, however, usually requires the manual intervention of the test operator. This thermal contrast is calculated according to Equation (11) below, which describes the relation of temperature *T*(*t*) at the observed time *t* and the temperature *T*(*t’*) at the reference time *t’* for each individual pixel of the record [28].
(11)$$\Delta T_{DAC}\left(t\right)=T\left(t\right)-\sqrt[{b}]{\frac{t^{\prime }}{t}}T\left(t^{\prime }\right)$$

Since the theoretical temperature decrease using the standard DAC method does not involve heat transfer by radiation, but only by conduction, this decrease is significantly lower than in reality. For this reason, the square root in Equation (11) was replaced by a power with the general parameter *b* representing the slope of the temperature decrease (12).
(12)$$\Delta T_{DAC}\left(t\right)=T\left(t\right)-{\left(\frac{t^{\prime }}{t}\right)}^{b}T\left(t^{\prime }\right)$$

This parameter was determined from experimental data based on the approximation of the cooling curve on the reference gauge for the thickness *z* = 2.30 mm, which represents the adhesive joint without any defect. The reference time for calculating the contrast is the time *t’* = 0.15 s, which corresponds to the thickness just behind the interface between the skin and the adhesive, and the value of the parameter *b* = 0.65. The whole calculation is performed for temperatures normalized to the maximum and minimum temperatures reached during the measurement from excitation to a steady state at the end of the measurement, over the entire measured area. Figure 13 shows the actual and theoretical cooling curves plotted using Equation (11) for three different thicknesses on the reference gauge (Figure 13a) and the DAC curves for all the thicknesses on the reference gauge (Figure 13b).

As in Figure 12, the following graph in Figure 14 shows the DAC profile normalized to the temperature range from the highest temperature (curve for *z* = 0.25 mm) to the lowest temperature (curve for *z* = 2.3 mm) at each measurement point. This graph shows the dimensionless contrast within the data obtained at a given time. A comparison of the two graphs shows that the DAC contrast calculation does not have a significant effect on the resulting image contrast.

Figure 15 shows a comparison between the raw thermogram image and the modified DAC-processed image. These images represent the same test area (end of the front wing spar) at the same time after excitation. In the first image, the reflection of the IR camera is visible in the lighter parts (the foam sandwich area, blue circle), and in the dark area (the area of the adhesive joint), a significant unevenness in the emissivity of the surface can be observed (green marking). The effect of the heating unevenness is marked by a red circle. These imperfections have been largely eliminated by the application of the DAC method.

### 2.3. Experimental System Description

A modular test system was designed and employed for the PT NDE method. The experimental setup can be seen in Figure 16. The basic hardware elements of this system consist of a FLIR A325SC bolometric uncooled IR camera (resolution 320 × 240 pixels; NETD < 50 mK; maximum scanning frequency, 60 Hz), an instrument unit equipped with a PC for test control and data recording and two flash lamps (2 × 1200 Ws).

The instrument unit works as the communication interface between the PC, IR camera and excitation lamps. It consists mainly of the cDAQ measuring and control system from National Instruments, equipped with analog output cards (for excitation lamp control) and digital input/output as well as other necessary auxiliary electronics.

A program was specifically created for this purpose in the LabView environment from National Instruments, and this was used to control the test’s processes and record the measured data. The processing and evaluation of the obtained data were performed in the MATLAB environment from MathWorks [29].

## 3. Results and Discussion

The following figures represent examples of the test results. Each individual image covers the tested area, which has a size of approx. 320 × 240 mm and, at a given resolution of the IR camera, represents a resolution of 1 × 1 mm for each pixel of the image. The final images were adjusted so that the grayscale range covers the entire range of the evaluated data (from white to black). The maximum temperature increase at the moment after excitation is up to 10 K. The temperature is not measured during the NDE process. Figure 17 and Figure 18 show a representative selection of the images on which the test results are demonstrated, with an explanation of the individual indications.

The area of the adhesive joint of the wing tip rib (right) and of the front wing spar cap on the lower wing skin side is pictured in Figure 17. The red lines mark the area of the front wing spar for the PT NDT evaluation process. The blue lines mark the area of the sandwich foam core reinforcements of the wing skin. From the point of view of the adhesive joint evaluation, the critical places are represented by the lighter shades of the corresponding color (Figure 17A), which can be interpreted as the voids in the adhesive joint. In Figure 17D, the indication marked represents an overflow of excessive adhesive outside the joint area. In Figure 17B, the indication marked is caused by the overlapping of the top layers of the skin and thus its localized doubling. In Figure 17C, the indication marked represents the region of resin accumulation at the point where foam core ends and sandwich skins are joined together.

Figure 18 represents the area of the front wing spar at the location of the fuel tank on the upper wing skin side. Again, the places with the voids in the adhesive layer are clearly visible (Figure 18A). The use of two different types of adhesives is visible in the bonded area (Figure 18B). This is due to the need for increased resistance to the influence of fuel in the fuel tank area (use of C-resin type). In Figure 18C, the indication marked represents the local reinforcement in the area of the fuel tank lid by the addition of one layer of fabric to the outer skin lay-up.

Due to the fact that no etalons with artificial defects were available for testing, an additional comparison of the NDE findings, with the actual condition of the adhesive joint at the failure area, was performed following the static strength test of the wing demonstrator. In the main wing spar area, the CFRP wing skin was removed to the depth of the adhesive joint. Figure 19 shows a comparison of the NDE findings with the actual condition of the adhesive joint. The upper part of the image contains an evaluated picture of the adhesive joint before the structural strength test. The lower part shows the condition of the adhesive joint after the strength test and the resulting wing damage. The individual colored circles mark the corresponding defects of the joint. The NDE measurement shows a good match with the actual condition of the joint. The smallest detected defect (marked in red) is about 3 mm in diameter. Due to the resolution of the IR camera, this dimension can be considered as the smallest detectable defect in the adhesive joint for a given test configuration. Except for defects in the adhesive layer, the use of two adhesives (EA9394 and C-resin) is clearly visible in the picture. In the lower half of the picture, two vertical cracks caused by a failure during the strength test of the wing demonstrator can be also seen.

Figure 20 compares the result of the NDE test with the actual condition of the adhesive joint in the end of the wing spar area. Areas without adhesive are clearly visible. This was caused by insufficient squeezing of the adhesive onto the whole area of the joint. Figure 21 shows a similar comparison of the adhesive joint of the skin and the rib at the location of the flap lever seating. The comparison shows a good match of the NDE with the actual condition of the adhesive joint. It is only in the upper left part of the image that nonglued areas indicated by the NDE method cannot be seen, which is probably due to the removal of the nonglued layer during the removal of the wing skin.

Figure 22 shows an overview image (a top and bottom view of the right wing) resulting from the test of the entire wing, composed of individual images. The tested area is limited to the adhesive joints only. The dark vertical areas represent the adhesive joints of the spar caps to the wing skin. The horizontal areas represent the adhesive joints of the ribs to the wing skin. Areas with insufficient adhesive coverage (brighter areas in the adhesive joints) can be seen along almost the entire length of the adhesive joints of the spar caps to the wing skin. Furthermore, the areas of overlap of the outer layers of the wing skin, or the reinforcements of the structural openings in the skin, are clearly visible. To give the reader an idea of the speed of the described method, we present an overview of the time required. The measurement itself lasted about 20 h, which could be shortened when using custom tooling for setting up the camera and flash lamps. Data processing took about 10 h. In total, there are 26 measurements on the top of the wing and 21 measurements on the bottom of the wing.

## 4. Conclusions

This article presents a possible method for testing the quality of adhesive joints in the composite thin-walled structures of light sports aircraft. It also presents a simplified procedure for determining the thermal properties of the materials used, which are not usually reported by the manufacturers of basic materials, using an IR camera. This measurement procedure is not intended to be an alternative to accurate laboratory methods. Its purpose is to give an idea of the properties that depend, among other things, on the production technology used by the manufacturer of the specific composite part under investigation. Prior to the actual testing of the adhesive joints of the wing, the procedure for the testing and evaluation of the measured data was set and verified on a reference gauge, which simulated the adhesive joint. These tests demonstrated a sufficient depth resolution of the test method. The generally achieved sensitivity of the method was confirmed, where it was possible to detect a defect of a size twice its depth below the surface of the inspected part [12,30,31]. The anticipated method sensitivity, which is sufficient to detect a defect of at least 3 mm in size, was confirmed. Although no verification was performed on the samples with artificial defects, the sensitivity was found to be quite sufficient when compared to known cases of catastrophic failure of adhesive joints in this category of aircraft. The authors of [32] describe the case of a nonglued area with a size of 500 mm leading to a catastrophic failure of the composite wing of the LSA-category aircraft. According to the authors of [33], the failure of an all-composite sailplane wing due to a 200 mm long, nonglued area is described. In addition, this technique makes it possible to assess the nature of defects in the adhesive and improve the production process. A comparison of the results of measurements performed on the adhesive joints of the tested wing showed a sufficient matching of the measured results with the actual state of the adhesive joint.

The NDE testing method described in this article demonstrated very good usability for the detection of the flows in the adhesive joints of the wing skins, ribs and the load-bearing spar structure made from thin CFRP. The achieved results are clearly interpretable and usable for both the qualitative and quantitative evaluation of adhesive joints. As well as defects in the adhesive layer, manufacturing technology defects and inaccuracies such as adhesive overflow, foam insert misalignments or composite layer overlaps are detectable.

Of course, it would be possible to improve the readability of the resulting images in particular by: (1) increasing the image sharpness and thus more accurately determining the shape of the defect, which can be achieved by using a higher-resolution IR camera; (2) reducing noise in images, which can be achieved by using an IR camera with greater sensitivity and stability or, alternatively, due to the relatively low initial heating, by the use of more powerful flash lamps. This would increase the signal-to-noise ratio.

## Figures and Tables

**Figure 1 materials-14-00533-f001:**
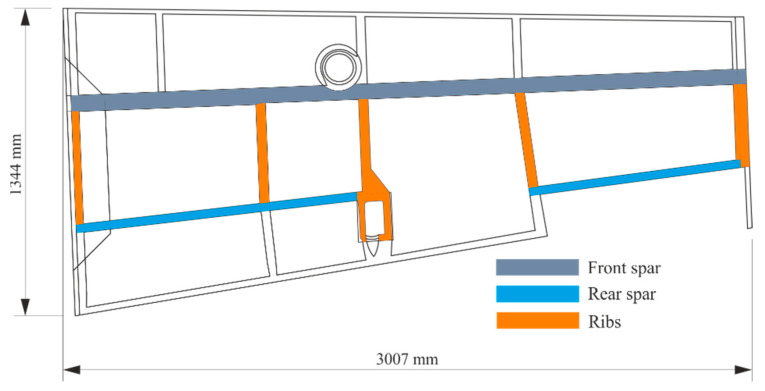
Composite wing configuration with the area of interest.

**Figure 2 materials-14-00533-f002:**
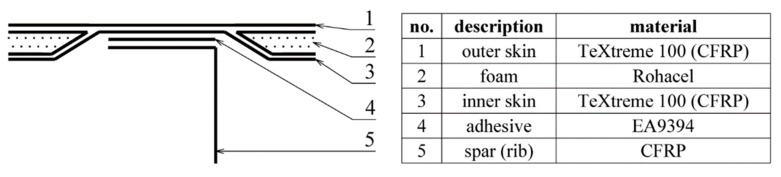
Bonded joint configuration.

**Figure 3 materials-14-00533-f003:**
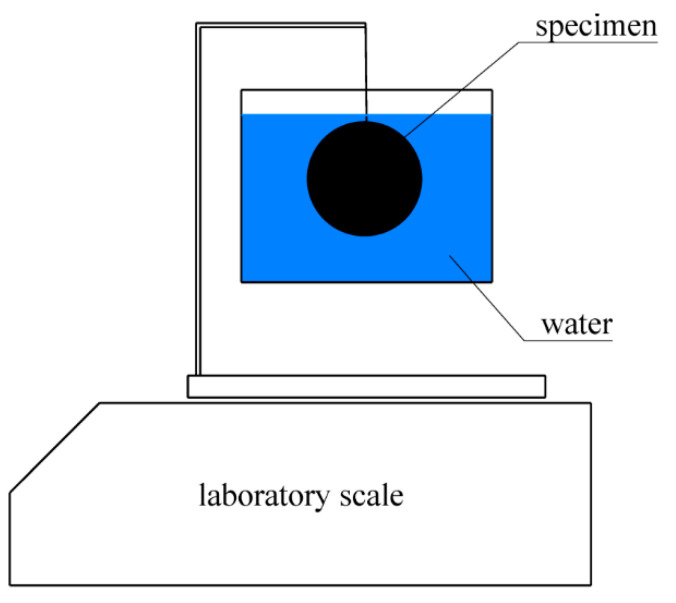
Density measurement configuration.

**Figure 4 materials-14-00533-f004:**
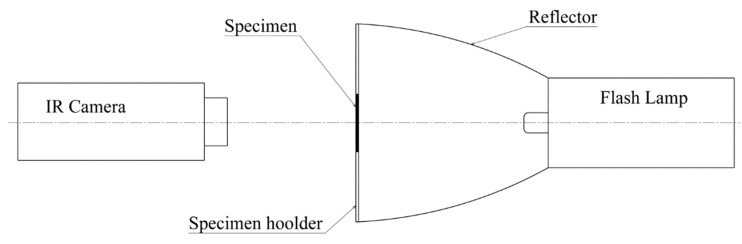
Thermal diffusivity and specific heat capacity measurement configuration.

**Figure 5 materials-14-00533-f005:**
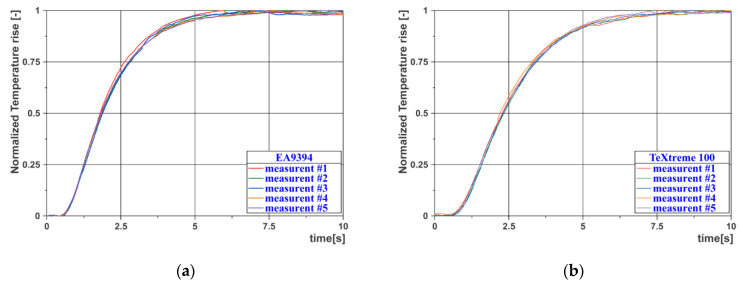
Measurement of thermal diffusivity—heating curves: (**a**) EA9394; (**b**) TeXtreme^®^ 100.

**Figure 6 materials-14-00533-f006:**
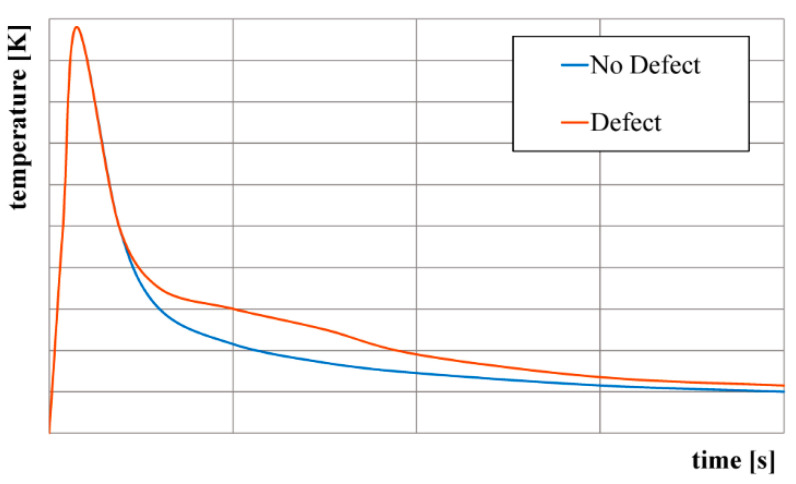
Surface cooling curves.

**Figure 7 materials-14-00533-f007:**
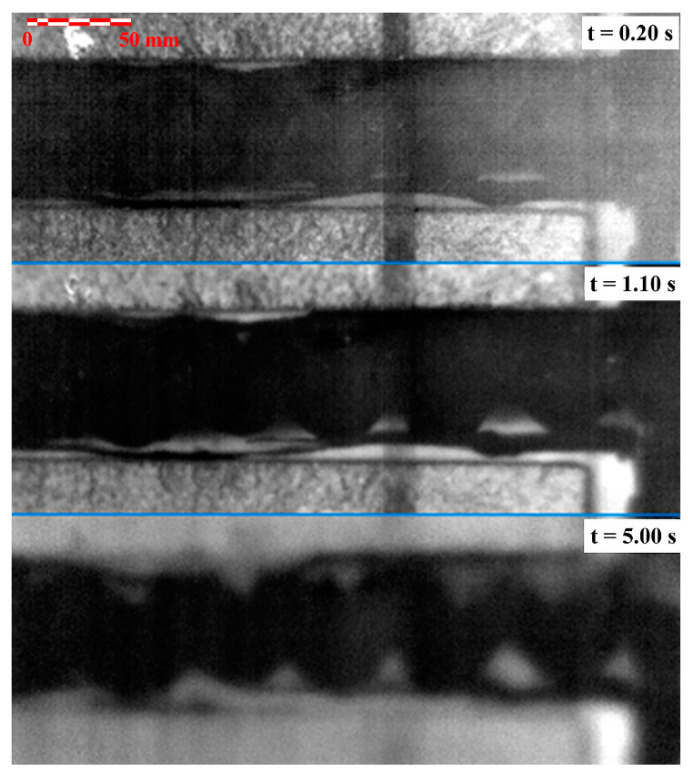
Sequence of the thermograms of the bonded joint.

**Figure 8 materials-14-00533-f008:**
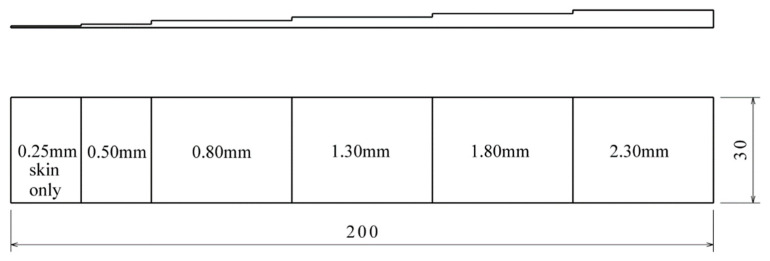
Dimensions of the reference gauge.

**Figure 9 materials-14-00533-f009:**
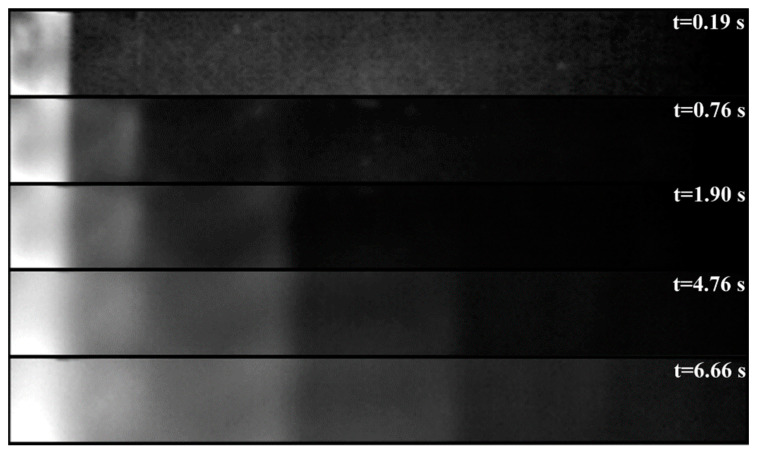
Thermogram sequence for reference gauge.

**Figure 10 materials-14-00533-f010:**
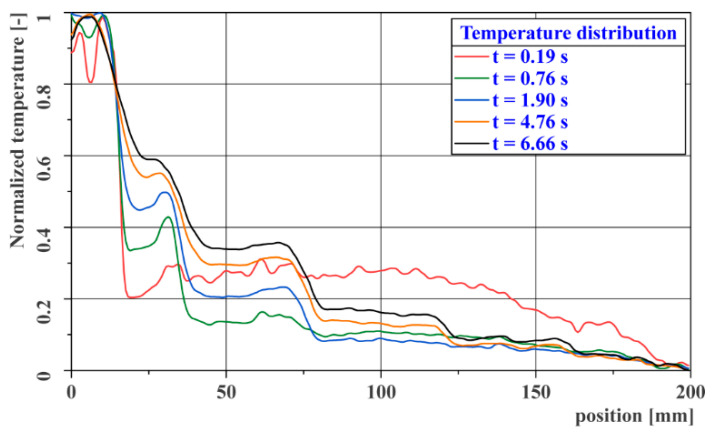
Course of temperature distribution along the longitudinal axis of the gauge.

**Figure 11 materials-14-00533-f011:**
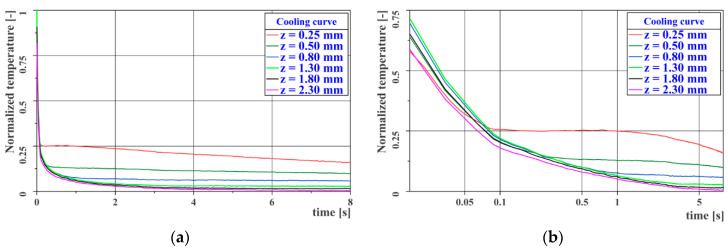
Cooling curves for the reference gauge: (**a**) linear time axis; (**b**) logarithmic time axis.

**Figure 12 materials-14-00533-f012:**
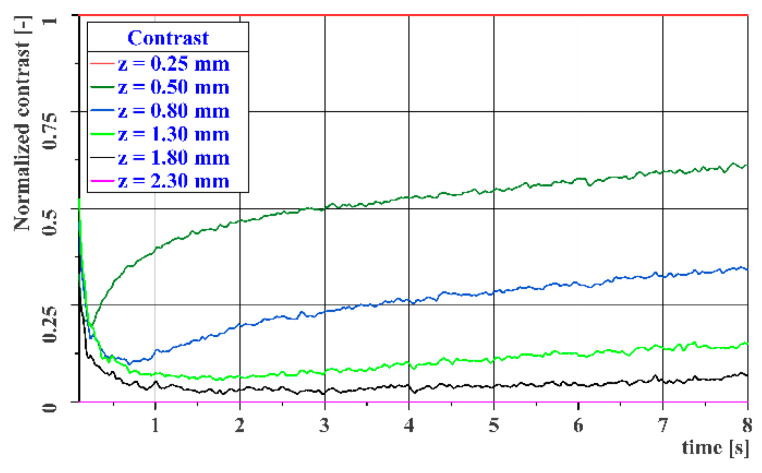
Image contrast curves for the reference gauge.

**Figure 13 materials-14-00533-f013:**
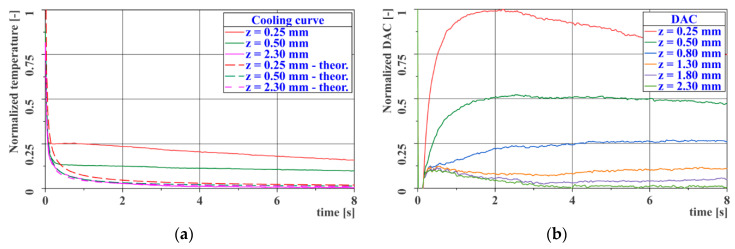
Cooling curves for the reference gauge: (**a**) linear time axis; (**b**) Differential Absolute Contrast (DAC) curves.

**Figure 14 materials-14-00533-f014:**
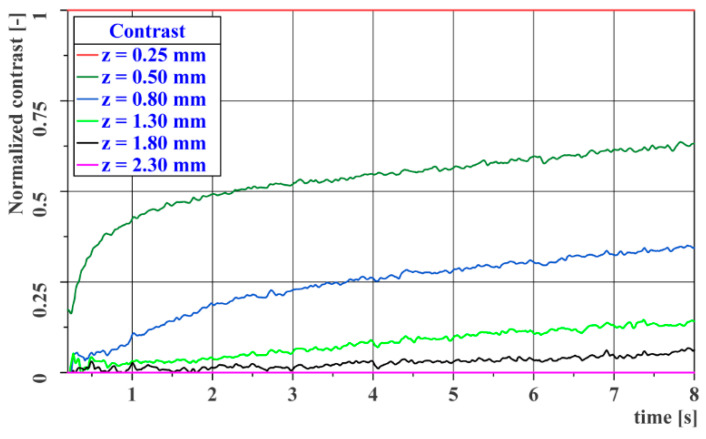
DAC image contrast curves for the reference gauge.

**Figure 15 materials-14-00533-f015:**
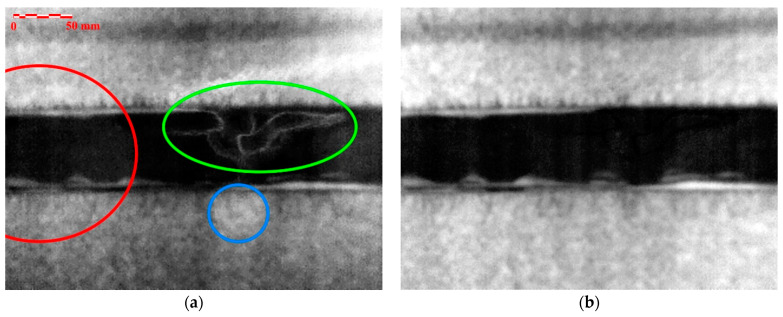
Comparison of the resulting images: (**a**) RAW thermal data; (**b**) modified DAC-processed data.

**Figure 16 materials-14-00533-f016:**
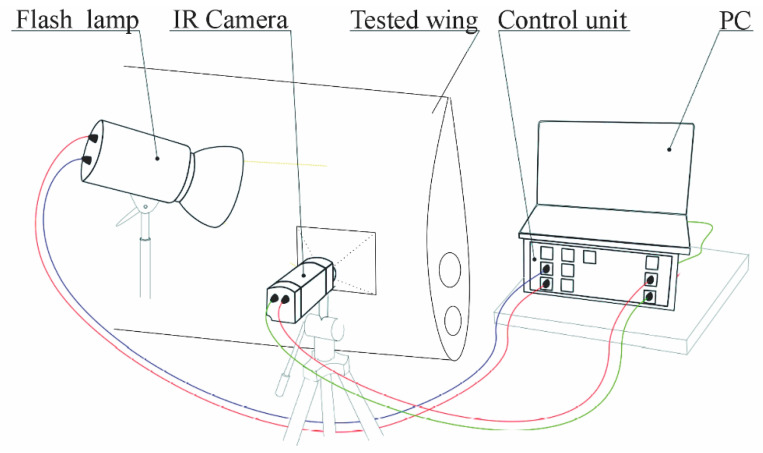
Pulse Thermography (PT) method experimental configuration.

**Figure 17 materials-14-00533-f017:**
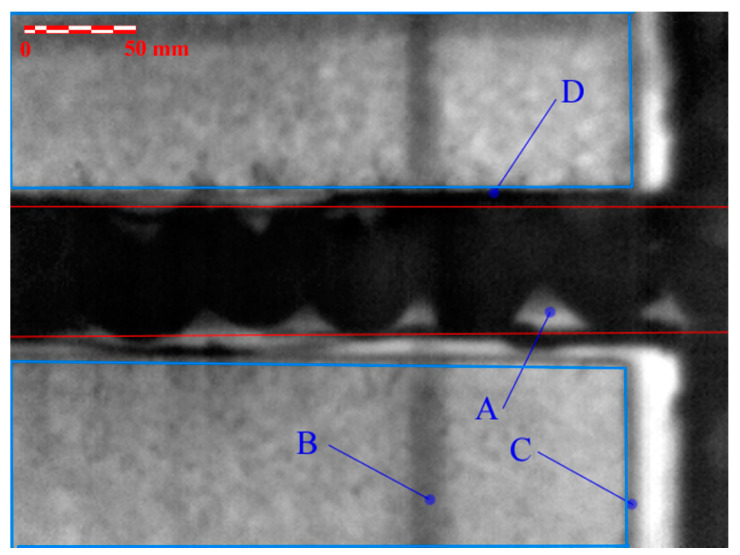
Results of the PT Nondestructive Evaluation (NDE) method: adhesive joint of the front wing spar and the wing tip rib on the lower wing side.

**Figure 18 materials-14-00533-f018:**
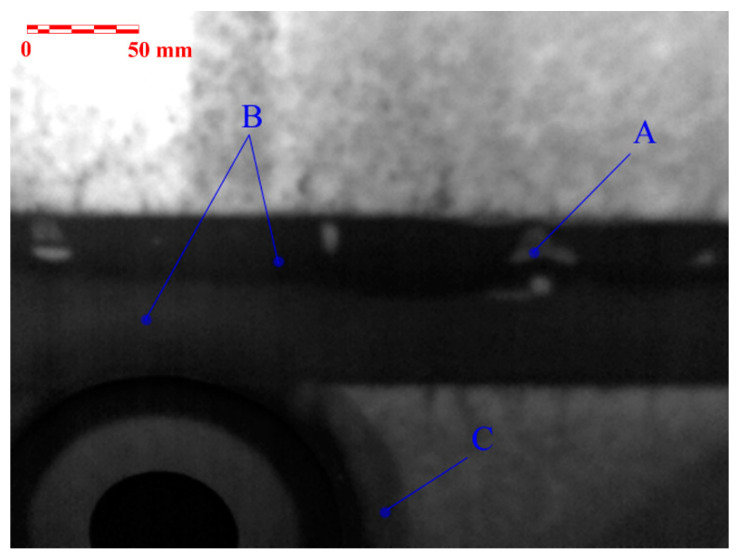
Results of the PT NDE method: adhesive joint of the front wing spar and the upper wing side in the fuel tank area.

**Figure 19 materials-14-00533-f019:**
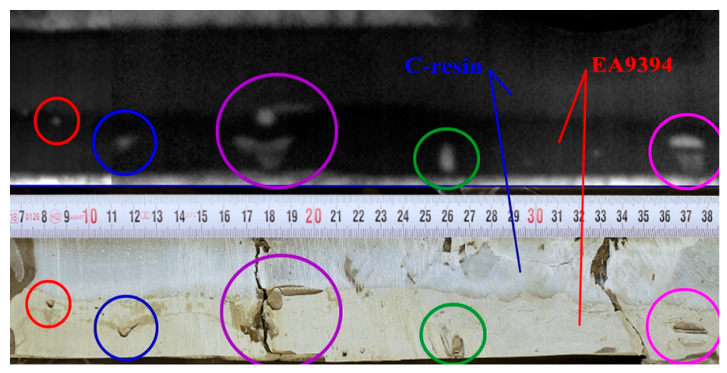
Comparison of NDE findings (**top**) with the actual condition of the adhesive joint (**bottom**).

**Figure 20 materials-14-00533-f020:**
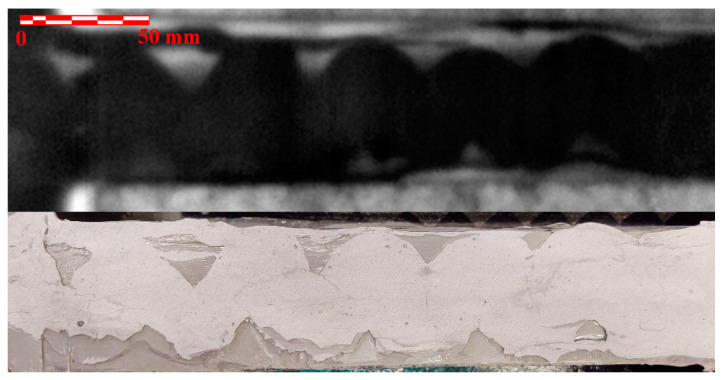
Comparison of NDE findings in the end of the wing spar area (**top**) with the actual condition of the adhesive joint (**bottom**).

**Figure 21 materials-14-00533-f021:**
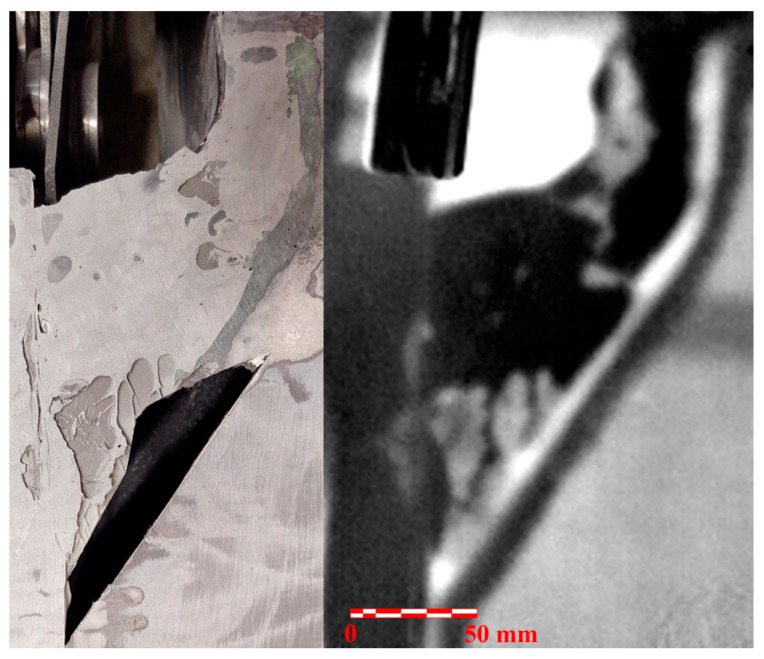
Comparison of NDE findings at the location of the flap lever seating (**right**) with the actual condition of the adhesive joint (**left**).

**Figure 22 materials-14-00533-f022:**
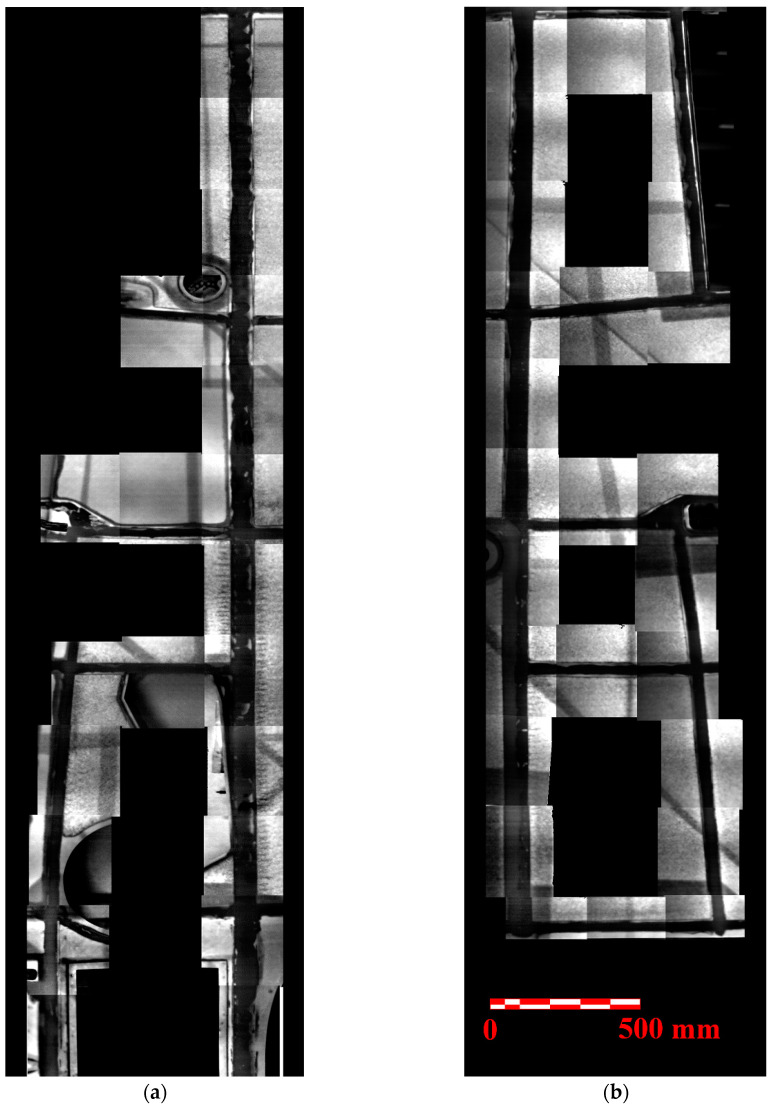
Results of the PT NDE method—overview image of the tested wing area: (**a**) lower wing side on the left; (**b**) upper wing side on the right.

**Table 1 materials-14-00533-t001:** Used materials’ densities.

Material	*m* (g)	Δ*m_w_* (g)	*ρ* (kg·m^−3^)
TeXtreme^®^ 100	7.235	4.650	1552
EA9394	4.187	3.000	1392
6061 T6	6.440	2.390	2687

**Table 2 materials-14-00533-t002:** Thermal diffusivity—EA9394.

Measurement No.	*z* (mm)	*t*_0.25_ (s)	*t*_0.50_ (s)	*t*_0.75_ (s)	*α*_0.5_ (m^2^s^−1^)	*α_corr_* (m^2^s^−1^)
1	1.745	1.253	1.797	2.622	2.352 × 10^−7^	2.118 × 10^−7^
2	1.745	1.254	1.853	2.789	2.281 × 10^−7^	2.227 × 10^−7^
3	1.745	1.289	1.885	2.816	2.242 × 10^−7^	2.140 × 10^−7^
4	1.745	1.246	1.848	2.760	2.287 × 10^−7^	2.220 × 10^−7^
5	1.745	1.245	1.821	2.755	2.320 × 10^−7^	2.250 × 10^−7^
Average	-	-	-	-	2.297 × 10^−7^	2.191 × 10^−7^
Stand. deviation	-	-	-	-	4.169 × 10^−9^	5.831 × 10^−9^

**Table 3 materials-14-00533-t003:** Thermal diffusivity—TeXtreme^®^ 100.

Measurement No.	*z* (mm)	*t*_0.25_ (s)	*t*_0.50_ (s)	*t*_0.75_ (s)	*α*_0.5_ (m^2^s^−1^)	*α_corr_* (m^2^s^−1^)
1	2.510	1.522	2.278	3.318	3.839 × 10^−7^	3.657 × 10^−7^
2	2.510	1.581	2.330	3.365	3.753 × 10^−7^	3.460 × 10^-7^
3	2.510	1.574	2.302	3.441	3.799 × 10^−7^	3.629 × 10^−7^
4	2.510	1.521	2.203	3.266	3.969 × 10^−7^	3.704 × 10^−7^
5	2.510	1.525	2.297	3.402	3.807 × 10^−7^	3.728 × 10^−7^
Average	-	-	-	-	3.833 × 10^−7^	3.636 × 10^−7^
Stand. deviation	-	-	-	-	8.195 × 10^−9^	1.054 × 10^−8^

**Table 4 materials-14-00533-t004:** Values of the temperature increase.

	Δ*T* (K)		
Measurement No.	EA9394	TeXtreme 100	6061 T6
1	7.809	6.660	8.867
2	7.744	6.655	8.874
3	7.785	6.738	8.874
4	7.832	6.597	8.930
5	7.861	6.547	8.959
average	7.806	6.639	8.901

**Table 5 materials-14-00533-t005:** Determined thermal properties.

Material	*ρ* (kg·m^−3^)	*α_corr_* (m^2^s^−1^)	*c_p_* (J·kg^−1^K^−1^)	*λ* (W·m^−1^K^−1^)	*e* (J·s^−0.5^m^−2^K^−1^)
EA9394	1392	2.191 × 10^−7^	1394	0.425	908.4
TeXtreme	1552	3.636 × 10^−7^	1022	0.557	957.0

## Data Availability

Data sharing is not applicable to this article.

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
