# Peer review of "Application of the Pulse Infrared Thermography Method for Nondestructive Evaluation of Composite Aircraft Adhesive Joints"

_materials, 2021, doi:10.3390/ma14030533_

Round 1

Reviewer 1 Report

The work presented in this paper is clear and interesting.
Defects in the adhesive joints have been examined by Pulse Thermography method.
The main issues are the small thickness of composite parts and inhomogeneity of the heat source and surface emissivity.

More literature references should be added in order to position the presented work and clarify its contribution. The novelty of the presented work should be clearly stated.

It is not clearly demonstrated that the modified DAC method reduces the effect of the heat source’s inhomogeneity and surface emissivity’s inhomogeneity.

Is it possible to estimate local adhesive thickness ?

Line 139: Figure 5, "cooling curves" should be "heating curves" ?

Author Response

The work presented in this paper is clear and interesting. Defects in the adhesive joints have been examined by Pulse Thermography method. The main issues are the small thickness of composite parts and inhomogeneity of the heat source and surface emissivity.

  1. More literature references should be added in order to position the presented work and clarify its contribution. The novelty of the presented work should be clearly stated.

CA: The Introduction part has been rewritten and expanded.

  1. It is not clearly demonstrated that the modified DAC method reduces the effect of the heat source’s inhomogeneity and surface emissivity’s inhomogeneity.

CA: The described defects were color-coded in Figure 15a and described in the text.

  1. Is it possible to estimate local adhesive thickness?

CA: This was not the subject of our research. We assume that with our experimental set-up this would be problematic.

  1. Line 139: Figure 5, "cooling curves" should be "heating curves"?

CA: Of course, this mistake was corrected.

Reviewer 2 Report

The paper deals with the application of the pulse infrared thermography non-destructive evaluation of composite adhesive joints of a small aircraft. The results are presented methodically so the idea is quite easy to follow. I appriciate the extensive description of the work. Although the paper is very good, I have some recommendations:

  • please use capital letters in the title of the paper except of prepositions and articles (that I think should be avoided in the title),
  • all abbreviations should be explained when they are used for the first time,
  • please re-check the the English (some missing articles, some missing prepositions and some typos),
  • some figures are closer to the text than others, please check formatting,
  • texts after equations starting with lower case letters should be aligned without tab, please correct,
  • please do not add a point after the number when reffering to a figure if it is not at the end of the sentence.

I have also a few remarks/recommendations:

  • You stated on lines 58 and 59, that ultrasonic and X-rays methods are not very suitable for detection of the defects in adhesive joints. You should also explain why (shortly).
  • I think, that the table 4 has wrong title, please modify.
  • Line 193 - ...layers no. 2,4,5... - shouldn't it be 3,4,5?
  • I am missing some geometric dimension scale in figures 7, 15, 17,18.
  • You haven't mentioned the temperature of the teststed specimen's surface after the excitation pulse, if possible, please add the information.
  • There are two vertical cracks in figure 19 (bottom), however only one is visible on the top part of the figure. You should explain this in the text.
  • Figure 22 looks really nice, good work. How long lasted the measurement and how long did it take to process the data? You can add some information about the time consumption of your method.

Author Response

The paper deals with the application of the pulse infrared thermography non-destructive evaluation of composite adhesive joints of a small aircraft. The results are presented methodically so the idea is quite easy to follow. I appriciate the extensive description of the work. Although the paper is very good, I have some recommendations:

  • please use capital letters in the title of the paper except of prepositions and articles (that I think should be avoided in the title),
  • all abbreviations should be explained when they are used for the first time,
  • please re-check the the English (some missing articles, some missing prepositions and some typos),
  • some figures are closer to the text than others, please check formatting,
  • texts after equations starting with lower case letters should be aligned without tab, please correct,
  • please do not add a point after the number when reffering to a figure if it is not at the end of the sentence.

CA: Thank you for all the recommendations, the article has been corrected.

I have also a few remarks/recommendations:

  1. You stated on lines 58 and 59, that ultrasonic and X-rays methods are not very suitable for detection of the defects in adhesive joints. You should also explain why (shortly).

CA: This was explained in the rewritten introduction part.

  1. I think, that the table 4 has wrong title, please modify.

CA: Table title has been corrected to “Values of temperature increase.”.

  1. Line 193 - ...layers no. 2,4,5... - shouldn't it be 3,4,5?

CA: Of course, it was corrected to 3, 4, 5.

  1. I am missing some geometric dimension scale in figures 7, 15, 17,18.

CA: Missing scale bars have been added to the respective figures.

  1. You haven't mentioned the temperature of the tested specimen's surface after the excitation pulse, if possible, please add the information.

CA: The following sentence has been added: “The maximum temperature increase at the moment after excitation is up to 10 K. The temperature is not measured during the NDE process.”.

  1. There are two vertical cracks in figure 19 (bottom), however only one is visible on the top part of the figure. You should explain this in the text.

CA: This is because the upper part of the image contains an evaluated picture of the adhesive joint before the structural strength test. The lower part shows the condition of the adhesive joint after the strength test and the resulting wing damage. The description in the relevant paragraph has been improved.

  1. Figure 22 looks really nice, good work. How long lasted the measurement and how long did it take to process the data? You can add some information about the time consumption of your method.

CA: The following sentence has been added: “To give reader an idea of the speed of the described method, we present an overview of the time required. The measurement itself lasted about 20 hours, this could be shortened when using custom tooling for setting up the camera and flash lights. Data processing took about 10 hours. In total, there are 26 measurements on the top of the wing and 21 measurements on the bottom of the wing.”.

Reviewer 3 Report

A well written and studied paper that could be almost ready for publication.

However, accuracy in state of the art and in some experimental descriptions needs checking. State of the art and results are not well presented and experimental data is not described in detail or properly to be more precise.

More importantly there are not presented other results from relevant NDT techniques to compare data. Comparative measurements are missing. There are not even results from reference knowledge-based samples.

In my point of view results are not convincing for suitability of the technique in a life-threatening application in case of evaluation failure, although a lot of research is going on from authors and other labs using the selected by the authors technique I don't find the results convincing for such an application. Higher sensitivity and resolution are much more critical parameters than fastness and cost for the described application.

I suggest to authors to find ways (even if it is not to be presented in this paper) to test other techniques as well and provide comparative data. Also reference known-defect samples.

Author Response

A well written and studied paper that could be almost ready for publication.

However, accuracy in state of the art and in some experimental descriptions needs checking. State of the art and results are not well presented and experimental data is not described in detail or properly to be more precise.

More importantly there are not presented other results from relevant NDT techniques to compare data. Comparative measurements are missing. There are not even results from reference knowledge-based samples.

In my point of view results are not convincing for suitability of the technique in a life-threatening application in case of evaluation failure, although a lot of research is going on from authors and other labs using the selected by the authors technique I don't find the results convincing for such an application. Higher sensitivity and resolution are much more critical parameters than fastness and cost for the described application.

CA: This was described in more detail in the introduction part. In short, no mandatory NDE control of the operated aircraft is prescribed in the LSA category, any simple and relatively inexpensive control method would represent an improvement of the current situation.

I suggest to authors to find ways (even if it is not to be presented in this paper) to test other techniques as well and provide comparative data. Also reference known-defect samples.

CA: Thank you for the recommendation, of course we assume further testing and comparison of the described method. Unfortunately, the current epidemiological situation does not allow us to cooperate effectively in this research.

Reviewer 4 Report

This paper investigates the use of the active infrared thermography as a non-destructive evaluation method for the test of composite aircraft adhesive joints and detection of defect therein. It is an interesting topic of field of composite material science. This manuscript could be worthy to be published after addressing the following issues.

  1. Overall the English is not satisfactory. Please carefully revised the grammar and typos errors.
  2. Please rewrite the sentence (line 113-115) how the temperature DTmax is defined?
  3. the errors of the values reported in the tables should be specified (or discussed in the text)
  4. the sensitivity of the method should be discussed and reported in the conclusion
  5. scale bars are missing on images Figs. 15, 17, 20-22.

Author Response

This paper investigates the use of the active infrared thermography as a non-destructive evaluation method for the test of composite aircraft adhesive joints and detection of defect therein. It is an interesting topic of field of composite material science. This manuscript could be worthy to be published after addressing the following issues.

  1. Overall the English is not satisfactory. Please carefully revised the grammar and typos errors.

CA: The article has been re-submitted for proofreading.

  1. Please rewrite the sentence (line 113-115) how the temperature DTmax is defined?

CA: The sentence has been rewritten to: The value of thermal diffusivity is calculated from the thickness of the sample z and the time t (measured from the excitation moment), when the temperature increase, ΔT, on the measured surface reaches a certain percentage of the maximum surface temperature increase ΔTmax.

  1. the errors of the values reported in the tables should be specified (or discussed in the text)

CA: These values have been described in the text using Equation (5).

  1. the sensitivity of the method should be discussed and reported in the conclusion

CA: In the conclusion, the sensitivity of the method was compared with known cases of a catastrophic failure of the adhesive joint of an all-composite aircraft of the same category.

  1. scale bars are missing on images Figs. 15, 17, 20-22.

CA: Missing scale bars have been added to the respective figures.

Round 2

Reviewer 3 Report

The paper has been positively enriched with important additions in state of the art, methodological procedure, analytical approach, conclusions and references; it is now considered ready for publication.

Author Response

Thank you for the inspiring comments and recommendations leading to the improvement of this article.